# Evaluating Tumor Size to Ki67 Proliferation Index Ratio for Optimizing Surgical Axillary Treatment Decisions in Breast Cancer Patients

**DOI:** 10.3390/cancers17050798

**Published:** 2025-02-26

**Authors:** Marco Pellicciaro, Marco Materazzo, Alice Bertolo, Federico Tacconi, Sebastiano Angelo Bastone, Francesco Calicchia, Denisa Eskiu, Enrica Toscano, Amir Sadri, Michele Treglia, Massimiliano Berretta, Benedetto Longo, Valerio Cervelli, Oreste Claudio Buonomo, Gianluca Vanni

**Affiliations:** 1Breast Unit Policlinico Tor Vergata, Department of Surgical Science, Tor Vergata University, Viale Oxford 81, 00133 Rome, Italy; marco.materazzo@ptvonline.it (M.M.); alibertolo00@gmail.com (A.B.); calicchiauniversita@gmail.com (F.C.); o.buonomo@inwind.it (O.C.B.); gianluca.vanni@ptvonline.it (G.V.); 2PhD Program in Applied Medical-Surgical Sciences, Department of Surgical Science, Tor Vergata University, 00133 Rome, Italy; 3Unit of Thoracic Surgery, Department of Surgical Sciences, Tor Vergata University, 00133 Rome, Italy; federico.tacconi@ptvonline.it (F.T.); angelo.bastone@ptvonline.it (S.A.B.); 4Faculty of Medicine, Università Cattolica Nostra Signora Del Buon Consiglio, 1000 Tirana, Albania; d.eskiu@prof.unizkm.al; 5Medical Oncology Unit, Department of Human Pathology “G. Barresi”, University of Messina, 98100 Messina, Italy; enricatoscano22@gmail.com; 6Plastic Surgery, Great Ormond Hospital for Children NHS Foundation Trust, London WC1N 3JH, UK; amir.sadri@gosh.nhs.uk; 7Department of Surgical Sciences, Tor Vergata University, 00133 Rome, Italy; michele.treglia@uniroma2.it; 8Laif (Laboratorio di Antropologia e Invecchiamento Forense), Sezione di Medicina Legale, Sicurezza Sociale e Tossicologia Forense, Tor Vergata University, 00133 Rome, Italy; 9Department of Clinical and Experimental Medicine, University of Messina, 98100 Messina, Italy; berrettama@gmail.com; 10Plastic and Reconstructive Surgery at Department of Surgical Science, Tor Vergata University, 00133 Rome, Italy; benedetto.longo@ptvonline.it (B.L.); valerio.cervelli@ptvonline.it (V.C.); 11Department of Health Science, University of Basilicata, Via Nazario Sauro, 85, 85100 Potenza, Italy

**Keywords:** breast cancer, axillary lymph node dissection, SNLB intraoperative evaluation, nodal burden disease: de-escalation

## Abstract

The role of axillary staging has changed in recent years, with a significant reduction in axillary dissection procedures. This has rendered intraoperative frozen section assessment obsolete. We recently observed the need for complete axillary staging in patients who are potential candidates for cyclin inhibitor therapy. The preoperative identification of patients with axillary lymph node disease and possible candidates for relevant therapies may be useful in selecting patients for whom intraoperative sentinel lymph node assessment could reduce the need for secondary surgery. The aim of our work is to evaluate the number of reoperations performed, with or without intraoperative lymph node assessment, and any predictors of axillary disease.

## 1. Introduction

Historically, the status of nodal involvement and sentinel lymph node metastasis have been the main predictive factors of oncological outcomes in invasive breast cancer and are considered to be of the utmost importance in determining the appropriate therapeutic approach [1]. Due to a decrease in its prognostic value, the approach to axillary surgery has become more conservative, with the aim of improving patients’ quality of life while maintaining the best oncological outcomes [1]. In recent years, several randomized clinical trials have demonstrated that even in the presence of moderate nodal involvement, local disease control can be achieved by omitting axillary lymph node dissection (ALND) [2,3,4]. According to the results of these randomized clinical trials, we have witnessed a de-escalation in the approach to axillary surgery, shifting from ALND to sentinel lymph node dissection (SLNB) and even its omission in selected cases [1]. This has been possible because, in addition to understanding the molecular characteristics of breast cancer, genomic tests have been developed to predict prognosis and response to adjuvant treatments [5]. However, although systemic therapy is tailored to the tumor’s intrinsic subtype, under some clinical conditions, the decision to add adjuvant treatment is still influenced by nodal status [4,5,6]. Despite advancements, there remains a lack of consensus on optimal criteria for axillary treatment decisions, particularly in the context of balancing treatment efficacy with quality of life considerations. This study seeks to fill this gap by evaluating the predictive value of the tumor size to Ki67 ratio, potentially offering a novel criterion for surgical decision-making. Therefore, despite the results of the above-mentioned studies, the outcome of sentinel lymph node biopsy alone may not be sufficient to select the most appropriate adjuvant treatment in some cases [7]. In these patients, a secondary surgery for axillary clearance will delay adjuvant systemic therapy and, therefore, impair the oncological outcome [8]. In order to reduce the risk of secondary surgery for axillary lymph node clearance, many centers conduct an intraoperative examination of the node using frozen sections, imprint cytology, or nucleic acid amplification [9]. However, due to the low probability of finding more than one metastatic lymph node in cN0 patients, performing ALND after intraoperative examination could be considered overtreatment without any relevant impact on adjuvant treatment.

The aims of our retrospective study are as follows: to compare patients subjected to SLNB with or without an intraoperative evaluation of the lymph node; to identify the predictive factors of higher burden disease in the axilla; and to determine potential candidates for adjuvant treatments influenced by nodal status, avoiding unnecessary ALND.

## 2. Materials and Methods

A single-center retrospective study of patients diagnosed with breast cancer who had undergone breast-conserving surgery between January 2019 and December 2023 was conducted by members of the Breast Unit of Policlinico Tor Vergata, Rome.

Preoperative diagnosis was achieved via fine needle aspiration, micro-biopsy, vacuum-assisted biopsy, or vacuum-assisted excision according to routine clinical practice, and the results were retrieved from preoperative histological examinations. Patients with cN0 were included; in cases where lymph nodes were suspected of oncological disease, they were evaluated only if a negative cytological examination had been performed. Patients then underwent SNLB with or without intraoperative examination.

All axillary procedures for lymph node staging were evaluated in this study. The removal of sentinel lymph nodes with or without complementary nodes (a maximum total of 5 lymph nodes) was classified as sentinel lymph node biopsy (SNLB); otherwise, it was considered axillary lymph node dissection (ALND).

According to the surgeon’s request, an intraoperative evaluation of the sentinel lymph node was performed by an expert pathologist using frozen sections or imprint cytology.

The patients were divided into groups according to the intraoperative evaluation of their sentinel lymph nodes: patients subjected to SLNB intraoperative evaluation (SLNB-IE) and patients in whom SLNB was evaluated during their final diagnostic evaluation (SLNB-DE). A sub-analysis of patients with T staging inferior to T2 was performed, and the participants were divided into groups SLNB-IE-T < 2 and SLNB-DE-T < 2 based on whether they underwent SLNB intraoperative evaluation. Histopathological characteristics were retrieved from the final pathological examination report, including the T stage, tumor size expressed in mm, nuclear grade, and breast cancer prognosis, along with predictive factors such as estrogen receptor (ER), progesterone receptor (PR), Ki67 proliferation index, and HER2 expression, as indicated by the recommendations of the 2018 ASCO/CAP [10]. Nodal status and the number of removed and metastatic lymph nodes were retrieved from the final pathological examination report and analyzed. The presence of more than 3 metastatic lymph nodes was considered to indicate high-burden axillary disease. All cases were discussed at a multidisciplinary meeting of the Breast Unit of Policlinico Tor Vergata. Data regarding re-operation for complete axillary clearance were retrieved from meeting reports and clinical notes.

The ratio of lesion dimension to ki67 proliferation index was considered an indicator of the time between cancer onset and surgery. Surgical time, defined as the time spent in the operating room, was collected from clinical notes and analyzed in this study. The patients had been previously informed and had signed consent for data retention for scientific purposes.

### Statistical Analysis

Data were recorded in an EXCEL database (Microsoft, Redmond, WA, USA). Categorical variables are reported as the mean and standard deviation. Continuous variables are presented as numbers and relative percentages and were analyzed using Student’s *t*-test. The Chi-squared test (or Fisher’s exact test, depending on group size) was used to analyze categorical dichotomous variables. If there were no dichotomous variables, the Monte Carlo test was adopted. All variables with a *p*-value of <0.05 were considered statistically significant. Multivariate logistic regression analysis was used to assess the effect of axillary burden disease. ROC curve analysis was performed to identify the predictive factors in the multivariate analysis, and the area under the curve (AUC) is reported. The sensitivity and specificity of the cut-off values were also reported. Statistical analyses were performed using the SPSS statistical package version 23.0 (SPSS Inc., Chicago, IL, USA).

## 3. Results

Of the initial sample of 741 patients, 551 (74.3%) underwent SLNB and were included in this study. In total, 333 (60.4%) patients underwent intraoperative evaluation of the sentinel lymph node, while 218 (39.6%) did not; instead, their sentinel lymph nodes were analyzed at the final histopathological examination. The mean maximum size of the lesions was 14.3 ± 8.2 mm. In 441 (80.0%) cases, the diagnosis was ductal carcinoma, and in 94 (17.1%) cases, it was lobular carcinoma. In 45 (8.2%) cases, lesions were multifocal, and in 13 (2.4%) cases, they were multicentric tumors. Of the 333 patients subjected to intraoperative evaluation of the sentinel lymph node, 44 (13.2%) presented with lymph node metastasis. Of the 108 (19.6%) patients with metastatic lymph nodes at the final pathological examination, ALND omission was performed in 53 (49.1%) cases.

At the definitive evaluation, 77 of 333 patients (23.1%) subjected to SLNB intraoperative evaluation (SLNB-IE) presented a metastatic sentinel lymph node versus 31 (14.8%) of the 218 patients belonging to the SLNB-DE group; the relative *p*-value was 0.008.

In seven (2.8%) cases, secondary surgery was performed in order to complete ALND in the SLNB-DE group versus two (0.6%) in the SLNB-IE group; the relative *p*-value was 0.032. Tumor T, evaluated using the Monte Carlo test, presented significantly higher staging in the SLNB-IE group compared with the SLNB-DE group (*p* = 0.010). Table 1 presents the baseline characteristics and staging between the groups with and without intraoperative evaluation of SLNB, highlighting significant differences in surgical time and T staging.

A significantly longer surgical time was observed in the SLNB-IE group at 103.51 ± 38.82 min versus 88.32 ± 30.89 min in the SLNB-DE group (*p*-value < 0.001).

### 3.1. Patients with T < 2

One hundred and twenty patients presented with T ≥ 2 and were excluded from the secondary analysis. Of the 431 patients included in the analysis, 244 (56.7%) underwent intraoperative evaluation of the sentinel lymph node (SLNB-IE-T < 2), whereas the intraoperative evaluation was not performed in 187 (43.3%) patients whose sentinel lymph nodes were analyzed at the final histopathological examination (SLNB-DE-T < 2).

Of the 244 patients subjected to intraoperative evaluation of the sentinel lymph node, 22 (9.1%) presented with lymph node metastasis. In 72 (16.6%) patients, SLNB metastasis was observed in the final pathological examination. Out of those 72, ALND was omitted in 42 (61.1%) cases. In five (1.2%) cases, a secondary surgery was performed to complete ALND, and a total of twenty-four (5.5%) cases were subjected to ALND. Of the 24 patients who underwent ALND, 11 (2.5%) presented with four or more metastatic lymph nodes. At the final pathological examination, 47 (19.2%) patients subjected to SLNB intraoperative evaluation (SLNB-IE-T < 2) presented with metastatic sentinel lymph nodes versus 25 (13.3%) patients in the SLNB-DE group; the relative *p*-value was 0.150.

In order to perform ALND, secondary surgery was performed on one (0.4%) patient in the SLNB-IE-T < 2 group and on four (2.1%) patients in the SLNB-DE-T < 2 group; the relative *p*-value was 0.171. Tumor staging and other baseline characteristics are displayed in Table 2.

A significantly longer surgical time was observed in the SLNB-IE group at 99.08 ± 35.98 min versus 88.74 ± 30.63 min in the SLNB-DE; the relative *p*-value was 0.002.

### 3.2. Predictive Factor of Tumor Burden in Axilla

In order to identify potential predictors of tumor burden in the axilla (more than three lymph nodes involved), a multivariate logistic regression was performed. For the multivariate analysis, the following factors were considered: age, Ki67 proliferation index, T staging, maximum diameter and a low ratio of lesion dimension to the Ki67 proliferation index. Among these, only lesion dimension (OR 1.678; 95%CI 1.019–2.145; WALD: 7.588; *p* = 0.006) and the ratio of lesion dimension to the Ki67 proliferation index (OR 0.08; 95%CI 0.011–0.141; WALD: 11.004; *p* = 0.001) remained independent predictive factors for a higher axillary tumor burden. In Table 3, the other parameters analyzed with relative *p* values are presented (Table 3).

A ROC curve analysis was performed in order to identify a cut-off value to predict the risk of tumor burden in the axilla, as shown in Figure 1. The AUC of this analysis was 0.873. A value of 0.425 (ratio of tumor dimension to Ki67 proliferation index) was identified as a predictor of tumor burden in the axilla, with a sensitivity of 78% and specificity of 87.5. Figure 1 illustrates the ROC curve analysis used to identify a cut-off value for predicting axillary tumor burden, with an AUC of 0.873, indicating a strong predictive model.

## 4. Discussion

Through this retrospective analysis, we found that in the era of axillary surgical de-escalation, intraoperative SLNB examination did not significantly reduce the risk of secondary surgery for axillary clearance in patients with early breast cancer T < 2 cN0. Intraoperative examination of SNLB in patients with T ≥ 2 cN0 could reduce the risk of a secondary surgery; however, in selected cases with a lower ratio of tumor dimension to the Ki67 proliferation index, it could be omitted due to the low risk of axillary disease. Our study introduces the tumor size to the Ki67 proliferation index ratio as a novel parameter in clinical decision-making. To our knowledge, no prior data have assessed this ratio’s prognostic value. While tumor size and Ki67 are typically considered separately, our findings suggest their combined assessment offers additional insight into tumor aggressiveness. This ratio could refine risk stratification and guide treatment decisions. Further validation in larger cohorts is needed to confirm its clinical relevance.

Over the last several decades, there has been continuous de-escalation of axillary surgery [1]. Even in cases with minimal-to-moderate axillary nodal burden, recent trials have demonstrated that local disease control can be achieved without ALND [2,3]. According to the results of these studies and the implementation of molecular and genomic tests, the role of axillary staging in adjuvant treatment has been reduced [4]. Even though adjuvant treatments are tailored according to tumor subtype and genomic results, under some clinical conditions, especially for HR+ breast cancers, the decision to include adjuvant systemic therapy is still influenced by the nodal disease burden [4,5]. We have witnessed a reduction in ALND, but the recent monarchE trial confirmed the benefit of CDK4/6 inhibitors in patients with four or more metastatic lymph nodes and HR+ breast cancer, interfering with the trend in axillary surgery de-escalation [7]. In this subgroup of patients, intraoperative examination of sentinel lymph nodes may reduce the risk of secondary surgery to complete axillary clearance. In our study, most patients who underwent secondary surgery were potential candidates for CDK4/6 inhibitors with an advanced T stage. In fact, when selecting patients with breast cancer classified as T2 or lower, this difference in terms of reintervention was no longer evident. In patients with T1 breast cancer, the incidence of metastatic lymph nodes can range between 10 and 15% [1]. In cN0 patients, the probability of having four or more metastatic lymph nodes after a positive sentinel node is low (ranging from 3.5% to 16% in published trials) [2,3,4,5,6,7,8]. In patients with T1 cN0 breast cancer, while waiting for the inclusion of SNLB omission in the guidelines, intraoperative evaluation of the sentinel lymph node could be safely omitted.

Most patients with HR+ breast cancer are at risk of having four or more positive lymph nodes, which are usually locally advanced and have higher T staging [8]. In this group of patients, the axillary cancer disease burden plays a fundamental role in the choice of systemic adjuvant treatment. Secondary surgery for axillary clearance will delay adjuvant systemic therapy and, therefore, impair the oncological outcome [8]. In patients who are potential candidates for CDK4/6 inhibitors and are cN0 with a higher ratio of tumor dimension to Ki67 proliferation index, intraoperative evaluation of SNLB should be performed in order to avoid secondary surgery and reduce the time between surgery and the initiation of adjuvant treatment. The identification of potential patients with lymph nodal disease without performing surgical axillary staging could allow for early therapy with CDK4/6 inhibitors, especially in younger patients, thereby reducing the risk of recurrence of the disease. However, this is only our supposition and should be demonstrated in randomized clinical trials.

The controversy surrounding axillary clearance is not yet resolved, but it could be partially resolved with the use of Ribociclib. In fact, the preliminary results of the NATALEE trial support the addition of Ribociclib to endocrine therapy in patients with HR + HER2−, including pN0 patients, overcoming the need to perform axillary lymph node dissection to determine eligibility for CDK4/6 inhibitors [10].

In our study, we found a correlation between the ratio of ki67 and tumor size with the disease burden of the axillary lymph node. As far as we know, this ratio has not been previously analyzed in the literature. Patients with a high value of this ratio have large tumors with a low proliferation index. In these patients, it seems that the risk of axillary disease burden is high, most likely because these are indolent diseases that arose a long time ago. The model in which slow-growing tumors are small and intrinsically less likely to metastasize has been confirmed in previous studies [11,12,13]. The relationship between tumor diameter and lymph node status in patients with invasive breast cancer is not linear [14].

Furthermore, the relationship between size and growth, expressed through the ratio of tumor diameter to proliferation index, has not been evaluated with regard to the axillary disease burden.

Large lesions with a low ki67 proliferation index take longer to grow, and the possibility of lymph node metastasis may be higher [15,16]. The higher the ki67 proliferation index, the faster the replication occurs and the higher the probability of having a mutation that favors distant metastasis and the attachment of tumor cells to distant sites [15,16,17,18]. In our opinion, time could play a fundamental role in local lymph node metastasis [16]. No parameters are currently available to determine when the disease begins [14]. Lesions that persist for an extended period can favor the transport of tumor cells into the lymphatic system and, therefore, local metastasis. Since there are no exact data on the time from tumor onset to surgery, the ratio of the tumor diameter to the ki67 proliferation index could be an estimate and possible prognostic factor for positive lymph nodes. We highlight that our conclusion is a supposition based on retrospective data; therefore, this parameter needs to be further explored with care and consideration, and its relationship with node metastasis should be prospectively demonstrated. A study considering the pathological mechanism of node metastasization, including adipose tissue, adipokines, or protein release, should be performed in order to identify potential pathways and relative inhibitors.

This study has several limitations. Firstly, it was a retrospective study, which could have introduced selection bias. Furthermore, the surgeon’s decision to perform sentinel lymph node intraoperative evaluation was influenced by tumor characteristics and their possible impact on adjuvant treatment. In fact, the patients subjected to intraoperative evaluation of SLN presented advanced T staging, and we tried to reduce the potential bias by only considering patients with T < 2 breast cancer in our analysis. Moreover, we believe that patients with T < 2 biological characteristics could also have influenced the surgeon’s decision. The follow-up period was relatively short, and the impacts of lymph nodal status, relative adjuvant treatments, and oncological outcomes were not the focus of this study and were therefore not evaluated; however, these factors will be analyzed in our prospective evaluation, which will be based on the results of this analysis.

## 5. Conclusions

Intraoperative evaluation of SNLB can be safely omitted in patients with T1 breast cancer. In patients with Tε2 breast cancer who are potential candidates for cyclin inhibitors and have cN0 with a higher ratio of tumor dimension to Ki67 proliferation index, intraoperative evaluation of SNLB can be conducted in order to avoid secondary surgery and reduce the time between surgery and the beginning of adjuvant treatment.

## Figures and Tables

**Figure 1 cancers-17-00798-f001:**
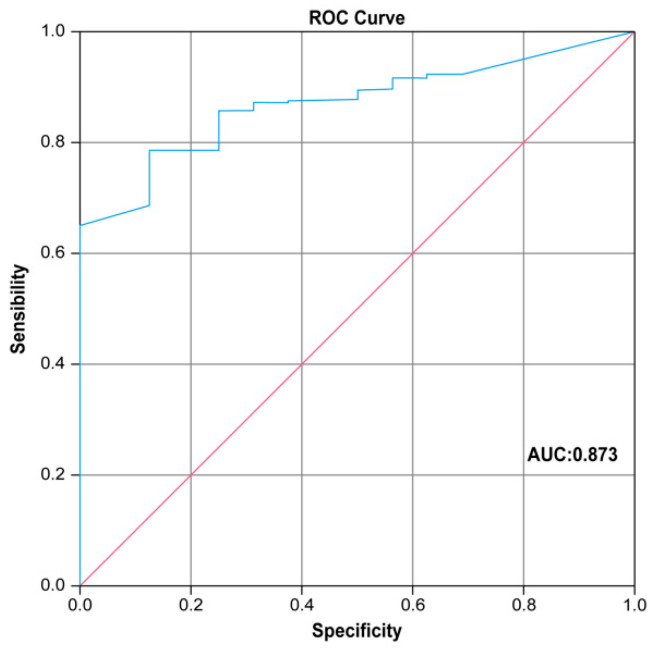
The ROC curve analysis conducted to identify a cut-off value to predict the risk of tumor burden in the axilla.

**Table 1 cancers-17-00798-t001:** Baseline characteristics and staging between groups with or without intraoperative evaluation of SLNB.

	SNLB-IE Group (n = 333)	SNLB-DE Group (n = 218)	*p*-Value
Age (years)	62.63 ± 12.35	64.07 ± 13.43	0.196
Tumor Diameter (mm)	15.10 ± 7.82	14.45 ± 7.82	0.050
Multifocal Lesion	26 (7.8%)	19 (8.7%)	0.751
Multicentric Lesion	12 (3.6%)	1(0.5%)	0.020
Histological Type			0.049
Ductal Carcinoma	261 (81.9%)	151 (69.2%)	
Lobular Carcinoma	46 (13.8%)	46 (21.1%)	
Others	24 (7.2%)	19 (8.7%)	
ER (%)	78.46 ± 28.09	79.30 ± 25.54	0.724
PR (%)	53.75 ± 38.84	57.83 ± 38.86	0.230
Ki67 (%)	20.33 ± 15.28	15.84 ± 12.91	0.001
HER2			0.747
Score 0	136 (40.8%)	96 (44.0%)	
Score 1	138 (41.4%)	87 (39.9%)	
Score 2	28 (8.4%)	12 (5.5%)	
Score 3	27 (8.1%)	18 (8.2%)	
Tumor Grading			0.132
Grade 1	78 (23.4%)	60 (27.5%)	
Grade 2	111 (33.3%)	67 (58.2%)	
Grade 3	98 (29.4%)	46 (21.1%)	
T Staging			0.008
Tis	4 (1.2%)	3 (1.4%)	
T1a	23 (6.9%)	17 (7.8%)	
T1b	62 (18.6%)	60 (27.5%)	
T1c	153 (45.9%)	101 (46.3%)	
T2	88 (26.4%)	31 (14.2%)	
T3	1 (0.3%)	0	
N Staging			0.003
N0	253 (75.9%)	183 (83.9%)	
N1a	64 (19.2%)	32 (14.6%)	
N1b	11 (3.3%)	2 (0.9%)	
N1c	4 (1.2%)	1(0.5%)	
N2	1 (0.3%)	0	
N3	0	0	

**Table 2 cancers-17-00798-t002:** Baseline characteristics and staging in patients with T < 2 between groups with or without intraoperative evaluation of SLNB.

	SNLB-IE-T < 2 Group (n = 244)	SNLB-DE T < 2 Group (n = 187)	*p*-Value
Age (years)	62.15 ± 11.68	63.03 ± 13.22	0.479
Tumor Diameter (mm)	12.33 ± 5.50	11.60 ± 9.82	0.346
Multifocal Lesion	18 (7.4%)	14 (7.5%)	0.106
Multicentric Lesion	8 (3.3%)	1(0.5%)	0.053
Histological Type			0.056
Ductal Carcinoma	192 (78.7%)	123 (65.7%)	
Lobular Carcinoma	34 (13.9%)	39 (20.8%)	
Others	17 (6.9%)	17 (9.09%)	
ER (%)	78.46 ± 25.59	79.83 ± 25.91	0.832
PR (%)	56.28 ± 38.54	56.54 ± 39.99	0.947
Ki67 (%)	17.78 ± 14.01	14.71 ± 13.53	0.026
HER2			0.520
Score 0	98 (40.2%)	82 (43.8%)	
Score 1	102 (41.8%)	71 (37.9%)	
Score 2	17 (6.9%)	12 (6.4%)	
Score 3	21 (8.6%)	10 (5.3%)	
Tumor Grading			0.438
Grade 1	72 (29.5%)	57 (30.5%)	
Grade 2	80 (32.8%)	47 (25.1%)	
Grade 3	60 (24.6%)	36 (19.2%)	
T Staging			0.378
Tis	4 (1.6%)	3 (1.6%)	
T1a	23 (9.4%)	17 (9.1%)	
T1b	62 (25.4%)	60 (24.5%)	
T1c	153 (62.7%)	101 (41.4%)	
T2	0	0	
T3	0	0	
N Staging			0.101
N0	192 (78.7%)	155 (82.9%)	
N1a	42 (17.2%)	26 (13.9%)	
N1b	5 (2.1%)	0	
N1c	2 (0.8%)	0	
N2	1 (0.4%)	0	
N3	0	0	

**Table 3 cancers-17-00798-t003:** Binary logistic regression for identifying predictive factors of a higher tumor burden in the axilla.

Multivariate			
Variables	OR	95%CI	*p*-Value
Age	0.973	0.930–1.018	0.241
Ki67 Proliferation Index	0.987	0.949–1.026	0.500
Tumor Dimension	1.678	1.019–2.145	0.006
Low-value Tumor Dimension/Ki67	0.008	0.001–0.141	0.001
T1a	0.334	0.530–1.160	0.999
T1b	0.465	1.019–1.117	0.980
T1c	0.502	0.048–5.223	0.564
T2	0.543	0.137–2.145	0.384

OR: odds ratio; CI: confidence interval.

## Data Availability

The data presented in this study are available upon request from the corresponding author subject to valid justification.

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
