# Peer review of "Evaluating Tumor Size to Ki67 Proliferation Index Ratio for Optimizing Surgical Axillary Treatment Decisions in Breast Cancer Patients"

_cancers, 2025, doi:10.3390/cancers17050798_

Round 1
Reviewer 1 Report
Comments and Suggestions for Authors
Different types of cancer were responsible for more than 8.2 million deaths worldwide in 2012. Moreover, the rapid increase in these numbers has been noted. Among them, breast cancer is one of the most frequent in females. Its early diagnosis causes the good prognosis success of therapy. However, node ablation is required in the tumour's advanced state. However, not total ablation can lead to its further surgery action and elongated time of chemotherapy treatment.
In the article entitled: Ratio between tumor size and Ki67 proliferation index as parameters for the coiche of surgical axillary treatments in breast cancer patients bring the prognostic, the authors take the above urgent problem for the first time into consideration.
The authors' main idea is to identify potential predictors of tumour burden in the axilla a multivariate logistic regression. Their analysis results disclose that axillary cancer burden plays a important role in the choice of systemic adjuvant treatments. Moreover, as has been correctly noticed in the article: In patients who are potential candidates for CDK4/6 inhibitor and are cN0 with higher ratio between tumor dimension and Ki67 proliferation index, intraoperative evaluation of SNLB should be performed in order to avoid secondary surgery and to reduce time between the surgery and the beginning of adjuvant treatment. In my opinion, the above is the main conclusion of the authors' studies.
The article is readable. However, following the author's idea to many unimportant numbers covering the main idea is challenging. I strongly recommend rewriting the manuscript. It would be nice if authors could reduce the typing mistakes. The mistake in the title is not acceptable.
The references are correctly selected and cited.
In the conclusion part, the importance of early chemotherapy treatment opportunities should be mentioned.
The article is interesting and valuable for some aroup of scientists, but in my opinion, it is far from the mainstream of Cancers journal. Therefore, I cannot recommend it for publication in its current form. Moreover, I suggest authors send it to more specific surgery journals such as Surgeries or Healthcare
Comments on the Quality of English LanguageI strongly recommend rewriting the manuscript. It would be nice if authors could reduce the typing mistakes. The mistake in the title is not acceptable.
Author Response
This manuscript titled as “Ratio between tumor size and Ki67 proliferation index as parameters for the coiche of surgical axillary treatments in breast cancer patients” focuses on the application of the ratio of tumor size to Ki67 proliferation index in axillary treatment for breast cancer patients undergoing surgery. Authors concluded that intraoperative evaluation of SNLB could be omitted, but in potential candidates for cyclin-inhibitor and cN0 with higher ratio between tumor dimension and Ki67 intraoperative evaluation of SNLB, could be considered in order to avoid secondary surgery. This research holds certain clinical significance and can provide extra ideas for axillary surgical decision-making. The overall research logic is clear and the structure is complete. Minor revision is recommended.
- This study is a retrospective study, which is prone to selection bias. Although the paper mentions this, it does not fully discuss how to minimize the impact of this bias on the results. In addition, the decision of whether to conduct intraoperative assessment by surgeons may be interfered by various factors, which will also affect the accuracy and reliability of the research results. Short - term follow - up is difficult to comprehensively evaluate the impact of lymph node status and adjuvant therapy on the long - term survival and recurrence of patients, and it is impossible to accurately determine the significance of the research results for the long - term prognosis of breast cancer patients. Moreover, this study was only carried out in a single center without multi - center verification, so the universality of the research results is questionable. The author needs to write a special chapter to discuss the limitations of this study.
Thank You for careful revision and your suggestions. We provide to expand limitations of the study, clinical characteristic and staging influenced surgical strategy. With multivariate analysis we try to reduce as much as possible the effects. The aim of our results is to stimulate colleagues and we want to found other center in other to expand the results as multicentric and prospective in order to avoid bias. IT will be interesting to investigate the impact on adjuvant treatment and long follow-up to understand the implication in term of overall survival and disease-free survival.
- The study believes that the ratio of tumor size to Ki67 proliferation index is related to axillary disease burden, but the exploration of its internal biological mechanism remains at the speculative level. It is recommended to supplement the discussion and summary of the relevant mechanisms to deeply explain the principle of this ratio in predicting axillary diseases.
Thank you we try to explain this correlation, and according to these results we start a prospective observational study, in collaboration with our biologist, in order to investigate the relations between biological characteristics, adipose tissue, secretoma and risk of nodes metastasis.
Reviewer 2 Report
Comments and Suggestions for Authors
This manuscript titled as “Ratio between tumor size and Ki67 proliferation index as parameters for the coiche of surgical axillary treatments in breast cancer patients” focuses on the application of the ratio of tumor size to Ki67 proliferation index in axillary treatment for breast cancer patients undergoing surgery. Authors concluded that intraoperative evaluation of SNLB could be omitted, but in potential candidates for cyclin-inhibitor and cN0 with higher ratio between tumor dimension and Ki67 intraoperative evaluation of SNLB, could be considered in order to avoid secondary surgery. This research holds certain clinical significance and can provide extra ideas for axillary surgical decision-making. The overall research logic is clear and the structure is complete. Minor revision is recommended.
- This study is a retrospective study, which is prone to selection bias. Although the paper mentions this, it does not fully discuss how to minimize the impact of this bias on the results. In addition, the decision of whether to conduct intraoperative assessment by surgeons may be interfered by various factors, which will also affect the accuracy and reliability of the research results. Short - term follow - up is difficult to comprehensively evaluate the impact of lymph node status and adjuvant therapy on the long - term survival and recurrence of patients, and it is impossible to accurately determine the significance of the research results for the long - term prognosis of breast cancer patients. Moreover, this study was only carried out in a single center without multi - center verification, so the universality of the research results is questionable. The author needs to write a special chapter to discuss the limitations of this study.
- The study believes that the ratio of tumor size to Ki67 proliferation index is related to axillary disease burden, but the exploration of its internal biological mechanism remains at the speculative level. It is recommended to supplement the discussion and summary of the relevant mechanisms to deeply explain the principle of this ratio in predicting axillary diseases.
Author Response
Different types of cancer were responsible for more than 8.2 million deaths worldwide in 2012. Moreover, the rapid increase in these numbers has been noted. Among them, breast cancer is one of the most frequent in females. Its early diagnosis causes the good prognosis success of therapy. However, node ablation is required in the tumour's advanced state. However, not total ablation can lead to its further surgery action and elongated time of chemotherapy treatment.
In the article entitled: Ratio between tumor size and Ki67 proliferation index as parameters for the coiche of surgical axillary treatments in breast cancer patients bring the prognostic, the authors take the above urgent problem for the first time into consideration.
The authors' main idea is to identify potential predictors of tumour burden in the axilla a multivariate logistic regression. Their analysis results disclose that axillary cancer burden plays a important role in the choice of systemic adjuvant treatments. Moreover, as has been correctly noticed in the article: In patients who are potential candidates for CDK4/6 inhibitor and are cN0 with higher ratio between tumor dimension and Ki67 proliferation index, intraoperative evaluation of SNLB should be performed in order to avoid secondary surgery and to reduce time between the surgery and the beginning of adjuvant treatment. In my opinion, the above is the main conclusion of the authors' studies.
The article is readable. However, following the author's idea to many unimportant numbers covering the main idea is challenging. I strongly recommend rewriting the manuscript. It would be nice if authors could reduce the typing mistakes. The mistake in the title is not acceptable.
Thank to your suggestion we provide to perform academic revision by MDPI revision system and correct the typing mistake.
The references are correctly selected and cited.
In the conclusion part, the importance of early chemotherapy treatment opportunities should be mentioned.
Thank you to the opportunity to discuss the importance of early chemotherapy treatment.
The article is interesting and valuable for some aroup of scientists, but in my opinion, it is far from the mainstream of Cancers journal. Therefore, I cannot recommend it for publication in its current form. Moreover, I suggest authors send it to more specific surgery journals such as Surgeries or Healthcare.
Thank you but we strongly believe our topic is not only a surgical problem and we hope that, with the revisions, we improve in quality the manuscript and it fulfils the criteria to be published in Cancers Journal. The publication of this manuscript could stimulate oncological community and not only the surgeons, because de-escalation should not be only surgical but multidisciplinary.
Round 2
Reviewer 1 Report
Comments and Suggestions for Authors
The authors made some efforts to improve their results and article. They also made the text correction.
The article can be accepted for publication.